# ADAPTIVE MASKING ENHANCES VISUAL GROUNDING

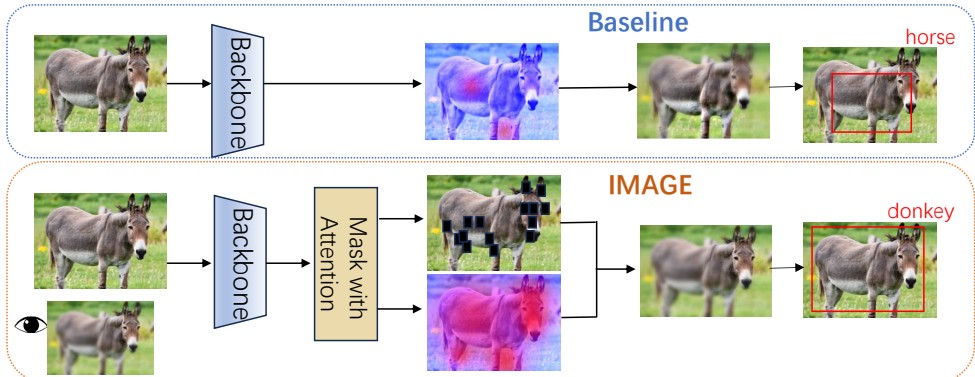

Figure 1: Our IMAGE method is inspired by human perception; by masking key details of objects, we encourage the model to learn more robust representations.

## ABSTRACT

In recent years, zero-shot and few-shot learning in visual grounding have garnered considerable attention, largely due to the success of large-scale vision-language pre-training on expansive datasets such as LAION-5B and DataComp-1B. However, the continuous expansion of these datasets presents significant challenges, particularly with respect to data availability and computational overhead, thus creating a bottleneck in the advancement of low-shot learning capabilities. In this paper, we propose a novel approach, **I**nterpretative **MA**sking with **G**aussian Radiation Mod**E**ling, aimed at enhancing vocabulary grounding in low-shot learning scenarios without necessitating an increase in dataset size. Drawing inspiration from cognitive science and the recent success of masked autoencoders (MAE), our method leverages adaptive masking on salient regions of the feature maps generated by the vision backbone. This enables the model to learn robust, generalized representations through the reconstruction of occluded information, thereby facilitating effective attention to both local and global features. We evaluate the efficacy of our approach on benchmark datasets, including COCO and ODinW, demonstrating its superior performance in zero-shot and few-shot tasks. Experimental results consistently show that IMAGE outperforms baseline models, achieving enhanced generalization and improved performance in low-shot scenarios. These findings highlight the potential of adaptive feature manipulation through attention mechanisms and Gaussian modeling as a promising alternative to approaches that rely on the continual scaling of dataset sizes for the advancement of zero-shot and few-shot learning.

## 1 INTRODUCTION

*"To see the world in a grain of sand,"* – William Blake, *Auguries of Innocence*

When observing an object, humans naturally focus on key details to grasp its essence. Masking these key features in visual tasks may encourage models to learn more robust representations, potentially

enhancing performance. Low-shot object grounding has gained significant attention due to its ability to reduce reliance on large labeled datasets. The capacity to ground and recognize novel objects with minimal examples is particularly valuable in applications like autonomous driving, where systems must handle rare or unseen situations with limited data Rezaei & Shahidi (2020). Additionally, low-shot grounding aids embodied AI in associating new concepts or objects within interactive environments with few labeled examples Varley et al. (2024). Recent vision-language models, such as CLIP Radford et al. (2021), have achieved notable success in bridging visual and textual modalities by leveraging large-scale pre-training. However, despite their strong performance, these models remain data-hungry, requiring substantial labeled data to adapt to new scenes. This reliance limits their utility in scenarios where data collection is challenging or impractical.

In visual grounding, recent efforts to enhance open-vocabulary detection have integrated textual prompts and multimodal fusion into object detection frameworks. Models like GLIP Li et al. (2022), YOLO-world Cheng et al. (2024), and Grounding DINO Liu et al. (2023) extend traditional detectors by incorporating language understanding, enabling object detection based on textual descriptions. While these approaches have advanced zero-shot grounding, they still demand extensive data to perform effectively. Furthermore, these models often struggle in complex scenes where visual cues are occluded or misaligned with textual descriptions.

These limitations highlight a critical issue: current multimodal models struggle to generalize from seen to unseen categories without explicit training examples. This challenge is compounded by their reliance on static visual cues and the lack of dynamic reasoning, as existing methods prioritize dataset expansion over teaching models to effectively "interpret" images. There are some methods such as Masked Autoencoder (MAE) He et al. (2022) and FLIP Li et al. (2023) attempt to improve the performance of a model by reconstructing the masked portion of the input data. However, this randomized masking approach suffers from poor interpretability, determinism and effect enhancement.

To address these limitations in a better way, we propose IMAGE, a novel method that introduces an adaptive masking strategy on features within the framework. Inspired by the human ability to infer missing information and focus attention dynamically, IMAGE mirrors cognitive processes in human reasoning. By deploying an adaptive mask scheme, IMAGE enables the model to learn more robust representations and focus on discriminative features.(eg. it makes more sense to identify cats by focusing on silhouette features rather than colors). In a word, IMAGE allows the model to learn how to "heed" objects rather than mechanically scanning and recognizing them.

We validate our method on datasets such as COCO Lin et al. (2014) and ODinW, and test it in both zero-Shot and few-Shot situations. Utilizing IMAGE's adaptive masking strategy, we achieve measurable improvements in both few-shot and zero-shot detection accuracy without significant computational overhead. Extrinsically, our method reduces the dependence on ever-larger datasets. Intrinsically, it provides a theoretical based way to empower existing detection models with robust learning and reasoning abilities. Our contributions are as follows:

- We introduce **IMAGE**, a novel adaptive masking framework that enhances low-shot visual grounding by enabling models to focus on important object features and improve reasoning capabilities, leading to more robust representations.

- We demonstrate theoretically and empirically that adaptive masking improves model robustness and generalization to unseen datasets, effectively addressing fundamental challenges in zero-shot and few-shot learning without relying on larger datasets.

- We provide empirical evidence on standard benchmarks showing that **IMAGE** outperforms baseline models and random masking strategies in low-shot settings, enhancing both few-shot and zero-shot performance with minimal computational overhead.

.

## 2 RELATED WORK

**Zero-Shot and Few-Shot Learning in Visual Grounding**   Low-shot learning, especially Zero-shot learning (ZSL), aims to recognize objects from unseen classes by leveraging knowledge transfer from seen classes Lampert et al. (2009); Farhadi et al. (2009); Socher et al. (2013). Early approaches

in ZSL for visual grounding focused on attribute-based methods and semantic embeddings to relate seen and unseen classes Akata et al. (2015); Xian et al. (2018). With the advent of large-scale vision-language models like CLIP Radford et al. (2021), recent works have utilized these pretrained models for zero-shot grounding tasks Gu et al. (2021); Li et al. (2022). However, these methods often rely on extensive datasets for pre-training and fine-tuning, limiting their scalability and practicality. Few-shot learning, on the other hand, seeks to learn new concepts from a small number of labeled examples Fei-Fei et al. (2006); Snell et al. (2017). In visual grounding, few-shot learning approaches have been developed to enhance generalization to new classes with limited annotated data Kang et al. (2019); Sun et al. (2021). Despite progress, many few-shot methods struggle with overfitting due to data scarcity and often require complex meta-learning frameworks Finn et al. (2017); Li et al. (2019).

**Attention Mechanisms and Masking Strategies in Vision Models**  Attention mechanisms have become integral in deep learning models for their ability to focus on relevant parts of the input data Bahdanau et al. (2015); Vaswani et al. (2017). In vision transformers, self-attention enables the modeling of global dependencies, enhancing feature representations Dosovitskiy et al. (2021); Liu et al. (2021). In the self-supervised learning area, masking parts of the input data has been an effective technique to improve feature representations. Methods like Masked Autoencoders (MAE) He et al. (2022) mask random patches of the input image and train the model to reconstruct them. BEiT Bao et al. (2021) extends this idea by using a tokenizer to create discrete tokens for masked patch prediction. However, these methods typically use random masking, which does not guide the model to focus on important features.

**Radiance Field Modeling and Gaussian Approaches**  Radiance fields have been employed in computer vision and graphics to model the way light interacts with surfaces, enabling high-fidelity scene reconstruction Mildenhall et al. (2020); Niemeyer et al. (2020). Neural Radiance Fields (NeRF) Mildenhall et al. (2020) represent scenes using continuous volumetric radiance fields parameterized by neural networks. Gaussian modeling of radiance fields allows for smooth representations and has been utilized in various applications Wang et al. (2021); Kim et al. (2022). Similarly, Zhou et al. Zhou et al. (2016) demonstrated that global average pooling enables CNNs to localize discriminative regions without explicit localization training. In our work, we employ a dynamic Gaussian modeling approach to represent the importance prior distribution of the feature map. This approach allows us to flexibly apply an adaptive mask to the feature map, instead of using rigid thresholding, thereby enhancing the model's focus on salient regions.

## 3 METHOD

**IMAGE** aims to enhance zero-shot and few-shot visual grounding without relying on large-scale datasets. Inspired by Masked Autoencoders (MAE), IMAGE leverages adaptive masking techniques that emphasize salient regions within an image's feature map, compelling the model to infer missing information and learn robust, generalized representations. As is shown in Fig. 2. IMAGE consists of two primary components: the **Importance Prior Generation Block** ($\theta_p$), which estimates the importance of image patches based on their relationships within the feature map, and the **Adaptive Mask Generation Block** ($\theta_m$), which creates adaptive masks guided by the importance prior to direct the model's attention during training. Given an input image, a pretrained Swin-Transformer backbone network processes it to produce hierarchical feature maps at multiple scales, denoted as $\{F_1, F_2, F_3, F_4\}$, where each feature map $F_i$ has dimensions $(B, C_i, H_i, W_i)$, representing batch size $B$, number of channels $C_i$, and spatial dimensions $H_i \times W_i$ at scale $i$. IMAGE applies adaptive masking on these feature maps, focusing the model's attention on the most relevant regions, thereby improving its reasoning capabilities and generalization performance.

### 3.1 IMPORTANCE PRIOR GENERATION

The first step in adaptive masking is to compute an *importance prior* that captures the relevance of each patch within a feature map. For each feature map $F_i$, we perform the following steps:

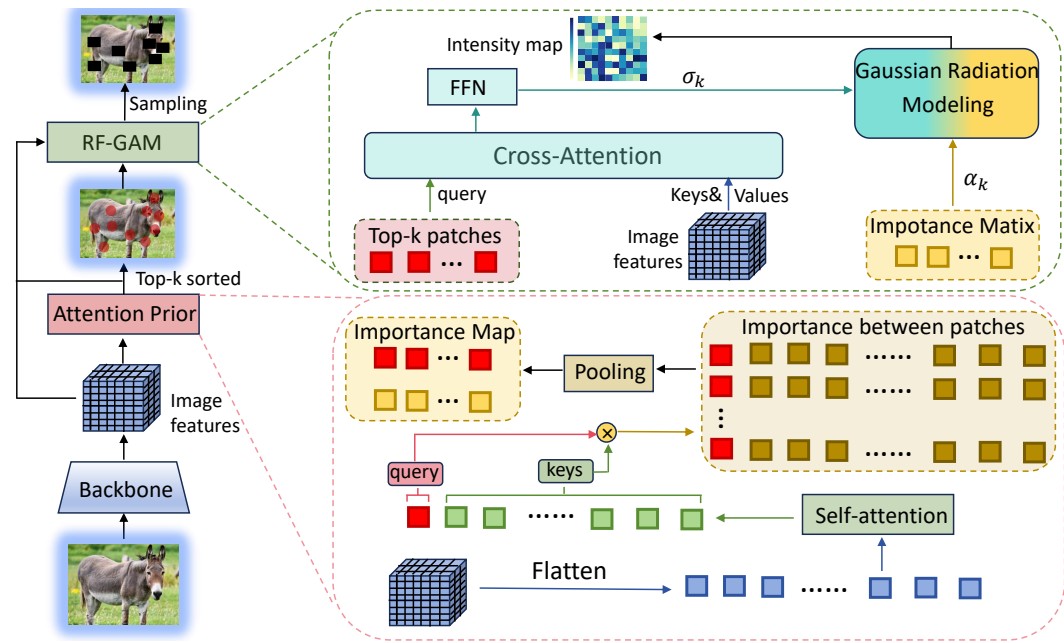

Figure 2: Pipeline of IMAGE model, consisting of two blocks: attention prior generation module and RF-GAM mask generation module.

**Self-Attention Encoding**  We reshape $F_i$ into a sequence of tokens and apply a self-attention mechanism to capture contextual relationships between image patches:

$$X_i = \text{Reshape}(F_i) \in \mathbb{R}^{B \times N_i \times C_i},$$
$$Z_i = X_i + \text{SelfAttention}(X_i),$$
$$T_i = Z_i + \text{FFN}(Z_i),$$

where $N_i = H_i \times W_i$ is the number of patches at scale $i$, and FFN denotes a feedforward network.

**Importance Prior Calculation**  To compute the importance of each patch, we calculate the correlation between each patch $p_j$ and all other patches in $T_i$:

$$S_j = p_j \times T_i^\top,$$

where $p_j \in \mathbb{R}^{B \times 1 \times C_i}$ is the feature vector of the $j$-th patch. We then average $S_j$ over all patches to obtain the importance score for patch $p_j$:

$$s_j = \text{AveragePooling}(S_j).$$

Repeating this for all patches yields the importance prior matrix $S_{\text{whole}} \in \mathbb{R}^{B \times N_i \times 1}$. We normalize the importance scores to ensure comparability:

$$\widetilde{S}_{\text{whole}} = \frac{S_{\text{whole}} - \min(S_{\text{whole}})}{\max(S_{\text{whole}}) - \min(S_{\text{whole}})}.$$

## 3.2 Adaptive Mask Generation

Using the importance prior $\widetilde{S}_{\text{whole}}$, we generate adaptive masks that obscure certain patches based on their importance. For the mask generation module, IMAGE proposes two mask generation strategies, corresponding to our adaptive mask and RF-GAM method respectively. The details are as follows:

**Threshold-Based Adaptive Masking**  We sort the patches based on their importance scores and designate the top $\rho_i\%$ as important regions for each scale $i$. Within these important regions, we

randomly select $\gamma\%$ of the patches to apply masking. For the remaining patches, we randomly mask patches to meet the desired masking ratio $\rho_i$. This strategy challenges the model to infer critical information from incomplete data while ensuring it has sufficient information to learn effectively.

**Radiance Field Gaussian Adaptive Masking (RF-GAM)** To implement a spatially aware masking strategy, we model the importance distribution using Gaussian radiance fields. For each feature map $F_i$, we select the top $K_i$ patches as radiation points based on their importance scores. For each radiation point $k$, we estimate its variance $\sigma_k^2$ by combining its feature vector $f_k$ with the cross-attention output $c_k$ and passing it through a feedforward network:

$$h_k = [f_k, c_k],$$
$$\sigma_k^2 = \text{ReLU}(\text{FFN}_\sigma(h_k)) + \epsilon,$$

where $\epsilon$ ensures numerical stability. The radiance intensity at each location $(x, y)$ is computed as:

$$I^{(b)}(x, y) = \sum_{k=1}^{K_i} \alpha_k^{(b)} \exp\left(-\frac{\|(x, y) - (x_k, y_k)\|^2}{2\sigma_k^{2(b)}}\right),$$

where $\alpha_k^{(b)}$ is the amplitude (importance score) of radiation point $k$. We determine masking thresholds based on the intensity distribution's mean $\mu^{(b)}$ and standard deviation $\sigma^{(b)}$:

$$T_{\text{hard}}^{(b)} = \mu^{(b)} + (\delta + k)\sigma^{(b)},$$
$$T_{\text{no-mask}}^{(b)} = \mu^{(b)} + (\delta - k)\sigma^{(b)},$$

with hyperparameters $\delta$ and $k$. The mask $M_i^{(b,p)}$ is defined as:

$$M_i^{(b,p)} = \begin{cases} 0, & \text{if } I^{(b)}(x, y) > T_{\text{hard}}^{(b)}, \\ 1, & \text{if } I^{(b)}(x, y) < T_{\text{no-mask}}^{(b)}, \\ 1 - \dfrac{I^{(b)}(x, y) - T_{\text{no-mask}}^{(b)}}{T_{\text{hard}}^{(b)} - T_{\text{no-mask}}^{(b)}}, & \text{otherwise.} \end{cases}$$

The final mask $M_i \in [0, 1]^{B \times N_i}$ is applied to the feature map:

$$F_i' = F_i \odot \text{Reshape}(M_i),$$

where $\odot$ denotes element-wise multiplication.

**Progressive Masking Strategy** To ensure effective learning, we introduce a progressive masking strategy that adjusts the masking ratio over the course of training:

- **Multi-Scale Masking**: Apply different masking ratios at different feature map scales. Lower-level feature maps retain more detail, while higher-level maps have higher masking ratios to focus on reasoning.
- **Dynamic Masking**: Gradually increase the masking ratio and mask strength during training. The hyperparameter $k$ in RF-GAM is adjusted per epoch:
$$k_{\text{epoch}} = k_0\left(1 - \frac{\text{epoch}}{E_{\text{total}}}\right),$$
where $k_0$ is the initial value, and $E_{\text{total}}$ is the total number of training epochs.

This progressive approach allows the model to adapt to increasing levels of difficulty, enhancing its ability to infer missing information and learn robust representations.

**Optimization and Learning Strategy**   To optimize the model effectively, we employ the following learning strategies:

- **Loss Functions**: We combine the standard contrastive loss used in vision-language alignment with the localization loss $L_{\text{localization}}$, weighted by $\beta$:

$$L_{\text{total}} = L_{\text{contrastive}} + \beta L_{\text{localization}}.$$

- **Training Schedule**: We adopt an asymptotic learning schedule, gradually increasing the masking difficulty as the model becomes more capable.

- **Hyperparameter Tuning**: Parameters such as the initial masking ratio, the rate of increase, and the thresholds in RF-GAM are tuned to balance the trade-off between learning from sufficient information and challenging the model.

By integrating the adaptive masking technique with a carefully designed optimization strategy, IM-AGE effectively enhances the model's ability to generalize from limited data without the need for scaling up dataset size.

### 3.3   THEORETICAL ANALYSIS

We provide a theoretical justification for how adaptive masking improves performance, drawing parallels to the principles of generalization.

**Assumption 1.** *The IAMGE model is trained on a dataset of image-text pairs $(x_i, t_i)$ drawn i.i.d. from an unknown joint distribution $\mathcal{D}$. Each image $x_i$ is encoded into a feature map $\mathcal{F}_i$, and the adaptive masking function generates a mask matrix $\mathcal{M}_i$ based on the importance prior learned from the feature map.*

**Assumption 2.** *The masking loss $L_{mask}$ is L-Lipschitz continuous with respect to the masked feature map $\mathcal{F}_i' = \mathcal{F}_i \odot \mathcal{M}_i$, where $\odot$ denotes element-wise multiplication.*

**Lemma 1.** *Let $\hat{y}_{ij}$ be the predicted similarity between the masked image feature embedding and the corresponding text embedding in a batch. Let $y_{ij}^*$ be the optimal similarity that minimizes the IMAGE loss $L_{IMAGE}$. Then, with probability at least $1 - \delta$, we have:*

$$|\hat{y}_{ij} - y_{ij}^*| \leq \frac{1}{\tau}\sqrt{\frac{\log(2/\delta)}{2N_{batch}}} + \frac{\beta L}{\tau},$$

*where $\tau$ is the temperature hyperparameter, $\beta$ is the masking loss weight, and $N_{batch}$ is the batch size.*

*Proof.* The first term arises from Hoeffding's inequality, which bounds the deviation between the empirical mean $\hat{y}_{ij}$ and the true expectation $y_{ij}^*$ of the similarity between masked image features and text embeddings. Since $0 \leq \hat{y}_{ij} \leq 1$, the deviation is bounded by:

$$\mathbb{P}\left(|\hat{y}_{ij} - \mathbb{E}[\hat{y}_{ij}]| \geq \epsilon\right) \leq 2\exp\left(-2N_{\text{batch}}\epsilon^2\right).$$

Solving for $\epsilon$ with probability $1 - \delta$ gives the first term.

The second term follows from the Lipschitz continuity of $L_{\text{mask}}$. By Assumption 2, for any two masked feature maps $\mathcal{F}_i'$ and $\mathcal{F}_i''$, we have:

$$|L_{\text{mask}}(\mathcal{F}_i') - L_{\text{mask}}(\mathcal{F}_i'')| \leq L\|\mathcal{F}_i' - \mathcal{F}_i''\|.$$

The deviation between the masked features of $\hat{y}$ and $y^*$ is bounded by their total variation distance, scaled by the Lipschitz constant and the loss weight $\beta$. Combining both terms completes the proof. □

**Theorem 1** (**IMAGE Generalization Bound**). *Let $f_\theta$ denote the IMAGE model with learned parameters $\theta$. Let $\hat{R}(f_\theta)$ and $R(f_\theta)$ denote its empirical and expected risks, respectively, on a downstream task. Then, with probability at least $1 - \delta$ over the training set, we have:*

$$R(f_\theta) \leq \hat{R}(f_\theta) + \mathcal{O}\left(\frac{1}{\tau}\sqrt{\frac{\log(1/\delta)}{N_{batch}}} + \frac{\beta L}{\tau}\right).$$

*Proof.* The empirical risk $\hat{R}(f_\theta)$ is an average over the predicted similarities $\hat{y}_{ij}$ for image-text pairs in the masked feature space. By Lemma 1 and applying a union bound over all $O(N_{\text{batch}}^2)$ pairs, each term $\hat{y}_{ij}$ concentrates around the optimal $y_{ij}^*$ with high probability. The total deviation is bounded by:

$$|\hat{R}(f_\theta) - R(f_\theta)| \leq \frac{1}{\tau}\sqrt{\frac{\log(2N_{\text{batch}}^2/\delta)}{2N_{\text{batch}}}} + \frac{\beta L}{\tau}.$$

Simplifying the logarithmic term and constants yields the stated bound. $\qquad\square$

**Discussion.** The theoretical analysis demonstrates that adaptive masking contributes to reducing the generalization error by forcing the model to learn robust representations from incomplete data. The generalization bound indicates that the error decreases with larger batch sizes $N_{\text{batch}}$ and appropriate choices of the temperature parameter $\tau$, masking loss weight $\beta$, and Lipschitz constant $L$. By adaptively masking key regions, the model is encouraged to develop stronger reasoning capabilities, which translates to improved performance in zero-shot and few-shot tasks.

# 4 EXPERIMENTS

## 4.1 EXPERIMENTAL SETUPS

**Dataset** We conduct experiments on the COCO and ODinW dataset. For training and evaluation in a close-set setting, we use the COCO 2017 dataset. The training set (train2017) contains approximately 118,000 images with 80 object categories, and the validation set (val2017) consists of about 5,000 images. To assess zero-shot detection capabilities, we utilize the ODinW datasets, specifically the ODinW_13 and ODinW_35 subsets. These datasets comprise images from various domains and contain object categories not present in the COCO training set, making them suitable for evaluating zero-shot performance. For few-shot experiments, we create subsets of the COCO train2017 dataset by randomly selecting 5%, 10%, 20%, and 30% of the data.

**Evaluation Metrics** We assess the performance of the proposed method using the following metrics: (1) **Average Precision (AP)**: Following the standard COCO evaluation protocol, we report the Average Precision at Intersection-over-Union (IoU) thresholds ranging from 0.5 to 0.95, denoted as AP@[0.5:0.95]. (2) **Zero-Shot Detection**: For the ODinW datasets, we use mean Average Precision (mAP) as the primary metric to evaluate the model's zero-shot detection performance. (3) **Few-Shot Performance**: To assess generalization in few-shot settings, we report AP on the COCO val2017 set, evaluating the model's ability to learn from limited data.

**Implementation Details** Our model is based on the Grounding DINO framework, incorporating a Swin-T backbone. The adaptive masking modules are integrated after the backbone's feature extraction stages, as described in Section 3. Different mask rates are applied to the four feature layers from the Swin-T backbone, with initial mask rates set to 20%, 30%, 40%, and 50%, respectively. In RF-GAM module, The parameter $k_0$ in the Gaussian modeling starts at 0.5 and decays smoothly to near zero over all epochs to facilitate progressive learning.

## 4.2 QUALITATIVE RESULTS

**Overall Performance** To assess the generalization capabilities of IMAGE, we compare their performance in zero-shot, close-set, and few-shot settings under the same number of training epochs. The experiment shows great improvement of our method in low-shot and close-set grounding tasks, As shown in Fig. 3. In particular, we also discuss the final results of these methods, shown in the table 1, where IMAGE still exhibits excellent performance.

In the close-set scenario, after training on the full COCO *train2017* dataset for 10 epochs, the RF-GAM model achieves an AP of 44.1% on the COCO *val2017* dataset, while the baseline model got 42.2% AP. In the few-shot scenario with 30% of the training data and 6 epochs, the model achieves an AP of 32.3%, which is close to the performance achieved by the baseline trained with the full dataset and outperforms the baseline model about 17% AP with 30% of the training data. This

| Datasets | Metric | Baseline | Random Mask | Adaptive Mask | RF-GAM |
|----------|--------|----------|-------------|---------------|--------|
| Close-set | COCO val2017 | 0.454 | 0.456 | 0.473 | **0.481 (+2.7%)** |
| Zero-shot | ODinW_13 | 0.208 | 0.190 | 0.235 | **0.251 (+4.3%)** |
| | ODinW_35 | 0.092 | 0.085 | 0.104 | **0.112 (+2.0%)** |
| Few-shot | COCO val2017 | 0.400 | 0.392 | 0.426 | **0.437 (+3.7%)** |

Table 1: Performance comparison across different datasets with percentage improvement in IMAGE in low-shot setting.

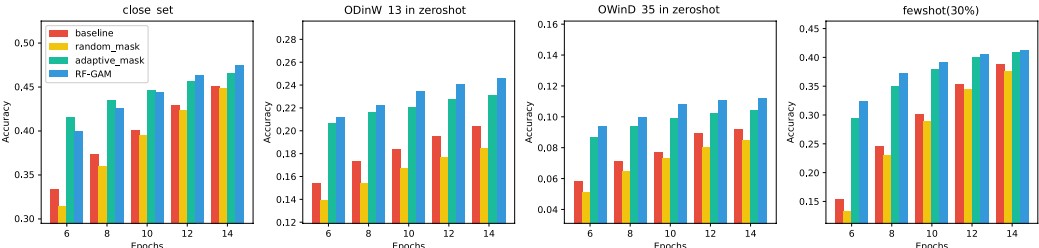

Figure 3: Scaling laws of our IMAGE model. With increased epochs, IMAGE achieves more accurate grounding AP across all four datasets and three settings.

highlights the efficiency of our method in low-data regimes and extraordinary robust representation learning .

For zero-shot evaluation, we test our models on the ODinW datasets, which contain categories not seen during training. As presented in Table 1, the RF-GAM model achieves an average AP of 25.1% on the ODinW_13 dataset, outperforming the baseline and random masking methods about 5% AP. This indicates that our adaptive masking strategies greatly enhance the model's ability to generalize to unseen categories, paving for the meta-learning in a new way.

**Few-Shot Training with Different Data Ratios**   We evaluate our models in few-shot scenarios by training them on varying proportions (5%, 10%, 20%, and 30%) of the COCO *train2017* dataset and testing on the COCO *val2017* dataset. This setup simulates situations with limited annotated data.

As shown in Fig. 4, our adaptive masking methods significantly improve performance in few-shot settings. For instance, with only 30% of the training data and after 6 epochs, the RF-GAM model achieves an AP of 32.3%, compared to 15.3% for the baseline model under the same conditions. In addition, RF-GAM with only 30% of the training data is already comparable in accuracy to the baseline with 100%. These demonstrates that RF-GAM enchants the model with incredible generalization ability and be able to learn effectively from limited data by focusing on critical features.

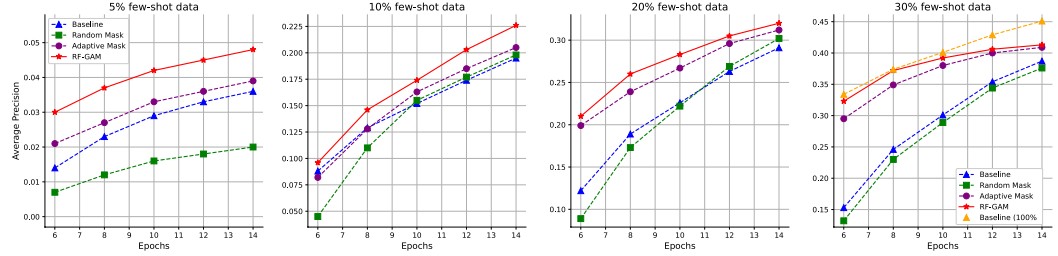

Figure 4: Comparison between IMAGE with other strategies in different few-shot ratios

**Impact of Different Mask Ratios**   To investigate the effect of different mask ratios on model performance, we experimented with various initial mask rates applied to different feature layers.

The mask ratios tested include [10%, 20%, 30%, 40%], [20%, 30%, 40%, 50%], and [30%, 40%, 50%, 60%].

The results in Table 2 indicate that the initial mask ratio of [20%, 30%, 40%, 50%] yields the best performance on most datasets, achieving the highest AP of 0.481 on the COCO *val2017* dataset, 0.112 on the ODinW_35 dataset, and 0.437 on the fewshot(30%) dataset. On the ODinW_13 dataset, the [30%, 40%, 50%, 60%] mask ratio gives the best performance with an AP of 0.253. These results suggest that a balanced masking strategy across feature layers generally enhances feature learning and overall model performance, but the optimal ratio may vary slightly depending on the dataset.

| Scale_masking_ratio | COCO val2017 | ODinW_13 | ODinW_35 | fewshot(30%) |
|---|---|---|---|---|
| [10%,20%,30%,40%] | 0.479 | 0.248 | 0.109 | 0.431 |
| [20%,30%,40%,50%] | **0.481** | 0.251 | **0.112** | **0.437** |
| [30%,40%,50%,60%] | 0.470 | **0.253** | 0.111 | 0.424 |

Table 2: IMAGE Results Across Different Datasets and Scale-masking Combinations

**The Performance of Methods in Different Occlusion Ratios** In the study of object grounding accuracy under partial occlusion, we compared our RF-GAM model, which utilizes adaptive masking and Gaussian dynamic modeling strategies, against a baseline model, random masking, and adaptive masking methods. AS shown in Fig. 5, as the occlusion rate increases from 0% to 80%, all models experience a decline in accuracy. However, RF-GAM consistently achieves the highest accuracy, particularly under higher occlusion rates, where its superiority becomes more evident. Even with 80% occlusion, RF-GAM still achieved **0.362** AP, far surpassing the baseline model **(0.136)** and outperforming both random and adaptive masking methods. This superiority can be attributed to the fact that the adaptive masking strategy based on Gaussian dynamic modeling in RF-GAM empowers the model to reason robustly from the residual image information.

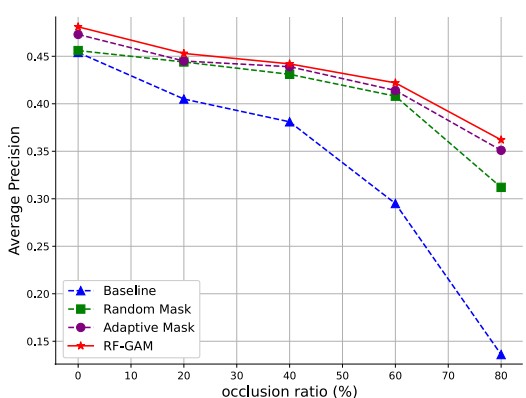

Figure 5: Results in different occlusion ratios on images across various methods.

## 4.3 ABLATION STUDIES

**Effectiveness of Importance Prior in Adaptive Masking** To assess the impact of incorporating importance priors in our adaptive masking strategy, we compare our model using adaptive masking strategy with a baseline and random masking where the masking is applied uniformly at random without considering patch importance. In random masking, patches are masked without regard to their significance in the feature map.

As shown in Table 1, the model utilizing the importance prior in adaptive masking methods demonstrates superior performance across all evaluation settings. Specifically, in the close-set scenario on COCO *val2017*, the model with importance prior achieves an AP of 47.3%, compared to 45.6%, 45.4% for the random masking and baseline respectively. In the zero-shot evaluation on the ODinW_13 dataset, the importance prior model attains an average AP of 23.5%, surpassing the baseline's 20.5% and 19% in random masking. For few-shot learning with 30% of the training data, the importance prior model achieves an AP of 42.6%, consistently outperforming the baseline's 40.0% and 39.2% in random masking. These results confirm that incorporating patch importance into the masking strategy effectively enhances feature learning by focusing on critical regions, leading to improved detection performance.

**Effectiveness of Gaussian Radiance Field Modeling**    We evaluate the contribution of RF-GAM by comparing it with the standard adaptive masking method that does not use radiance field modeling. The standard adaptive masking applies masking based on patch importance but without modeling the spatial distribution of importance using Gaussian functions.

As presented in Table 1, the RF-GAM method consistently outperforms the standard adaptive masking method across all scenarios. In the close-set evaluation on COCO *val2017*, RF-GAM achieves an AP of 44.1%, compared to 43.7% for the standard adaptive mask. In zero-shot detection on ODinW_13, RF-GAM attains an average AP of 25.1%, exceeding the standard method's 23.5%. In the few-shot setting with 20% training data and 6 epochs, RF-GAM achieves an AP of 32.3%, higher than the standard adaptive mask's 29.0%. These improvements indicate that modeling the importance distribution using Gaussian radiance fields allows for more nuanced and effective masking, enhancing the model's ability to learn salient features.

| Settings | Close-set (COCO) | Few-shot (COCO) | Zero-shot (ODinW_13/35) |
|---|---|---|---|
| non-progressive | 0.476 | 0.426 | 0.235 / 0.110 |
| progressive | **0.481** | **0.437** | **0.251 / 0.112** |

Table 3: The comparison of non-progressive and progressive training across datasets.

**Effectiveness of Progressive Training Strategy**    To determine the impact of the progressive training strategy, we conduct experiments where the parameter $k$ in the RF-GAM method is held constant, effectively removing the progressive aspect. In the standard RF-GAM, $k$ starts at an initial value (e.g., 0.5) and decays to near zero over the training epochs to facilitate gradual learning. By fixing $k$, we assess whether the progressive adjustment contributes to performance gains.

The results in Table 3 reveal that the progressive training strategy significantly enhances model performance. Without progressive $k$ decay, the IMAGE model achieves an AP of 47.6% on COCO *val2017*, which is slightly lower than the 48.1% achieved with the progressive strategy. Similarly, in zero-shot detection on ODinW_13, the non-progressive model attains an average AP of 23.5%, compared to 25.1% with progressive training. In the few-shot scenario with 30% data, the non-progressive model achieves an AP of 42.6%, lower than the 43.7% with progressive $k$ decay. These results suggest that gradually reducing $k$ during training helps the model adaptively adjust the masking intensity, promoting better feature learning and generalization.

## 5    CONCLUSION

In this paper, we introduced **IMAGE** (**I**nterpretative **MA**sking with **G**aussian Radiation Mod**E**ling), a novel approach designed to enhance zero-shot and few-shot visual grounding without the need for enlarging dataset sizes. Inspired by cognitive science and the success of Masked Autoencoders (MAE), our method employs adaptive masking on salient regions of the feature maps generated by the vision backbone, compelling the model to reconstruct occluded information and thereby learn robust, generalized representations that effectively attend to both local and global features. Evaluated on benchmark datasets including COCO and ODinW, IMAGE consistently outperforms baseline models, demonstrating superior performance in zero-shot and few-shot tasks. These findings underscore the potential of adaptive feature manipulation through attention mechanisms and Gaussian modeling as a promising alternative to methods relying on dataset scaling for advancing low-shot learning capabilities.

The challenges posed by complex real-world visual data, such as severe occlusions and missing key object features, present opportunities to further enhance our approach. Future work could focus on integrating more sophisticated data augmentation techniques or incorporating multimodal data—such as depth information or temporal cues—to improve the model's ability to generalize from incomplete visual inputs. Additionally, applying our adaptive masking strategy to other areas like video understanding or 3D vision may extend its benefits. A deeper investigation into the interplay between adaptive masking, attention mechanisms, and Gaussian modeling may provide valuable insights, potentially leading to further advancements in zero-shot and few-shot learning across various domains.

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
