# OpenReview forum: "Adaptive Masking Enhances Visual Grounding"
_ICLR.cc/2025/Conference — ICLR 2025 Conference Withdrawn Submission_

### Official Review · Reviewer_x81c · 2024-10-17

**Soundness:** 3
**Presentation:** 3
**Contribution:** 3
**Rating:** 6
**Confidence:** 4

**Summary:**

In this paper, the authors focus on the generalizable and robust feature representation issue in object detection task, and propose a novel masking strategy namely IMAGE (Interpretative MAsking with Gaussian Radiation ModEling). Specifically, IMAGE first leverages an attention-based importance prior module to measure the importance of each image patch, which can be seen as ranking measure of image patches. Then, IMAGE proposes Radiance Field Gaussian Adaptive Masking to model the importance distribution using Gaussian radiance fields, and then selects the masked region via thresholding. Experimental results show that IMAGE can significantly improve the generalization ability of object detectors with fast adaptation (e.g., 10 epochs).

**Strengths:**

1. The motivation of masking only a few key details in the image is intuitive and interesting. Such method may be helpful to enforce the network to capture generalizable information to recognize objects, such that improving the generalization ability.

2. The theoretical analysis is convincing.

3. The experimental results show the effectiveness of the proposed method.

4. The quantitative analysis is extensive and also convincing.

**Weaknesses:**

1. Though the experimental is promising, my main concern is that, 10-epoch training on COCO2017 (stated in sec. 4.1) is not enough for Grounding DINO fine-tuning [1]. Current results may only show that IMAGE largely improves the generalization ability in fast adaptation scenarios. The authors could follow the original fine-tuning setting (e.g., 50 epochs) to verify the effectiveness of IMAGE.

2. Adding some qualitative results to clearly demonstrate the heatmap of patch-level importance / masked results with different masking strategies can improve the readability of this manuscript.


[1] Grounding DINO: Marrying DINO with Grounded Pre-Training for Open-Set Object Detection, ECCV 2024.

**Questions:**

1. Is the RF-GAM module still used during inference? Intuitively, masking strategy for object detection tasks should be model-agnostic and can be removed during inference. The authors could clarify this point in the implementation detail section.

---

### Official Review · Reviewer_EDy6 · 2024-10-30

**Soundness:** 3
**Presentation:** 3
**Contribution:** 2
**Rating:** 3
**Confidence:** 4

**Summary:**

This paper introduces IMAGE (Interpretative MAsking with Gaussian Radiation ModEling), a novel approach designed to enhance zero-shot and few-shot visual grounding without the need for enlarging dataset sizes. The proposed scheme outperforms baseline models and random masking strategies in low-shot settings, enhancing both few-shot and zero-shot performance with minimal computational overhead.

**Strengths:**

1. This paper proposes a novel adaptive masking framework that enhances low-shot visual grounding by enabling models to focus on important object features and improve reasoning capabilities, leading to more robust representations.
2. It demonstrates theoretically and empirically that adaptive masking improves model robustness and generalization to unseen datasets.
3. Evaluated on benchmark datasets including COCO and ODinW, IMAGE consistently outperforms baseline models, demonstrating superior performance in zero-shot and few-shot tasks.
4. These findings underscore the potential of adaptive feature manipulation through attention mechanisms and Gaussian modeling as a promising alternative to methods relying on dataset scaling for advancing low-shot learning capabilities.

**Weaknesses:**

1. The novelty of this paper is insufficient, as the approach primarily involves excluding the most recognizable regions, which is common in object detection and segmentation. It would be beneficial to compare with other methods that have the same goal, e.g., Random Erase, to highlight differences.
2. The method of finding Top-k patches is somewhat unreasonable, as the detected patches may correspond to the background rather than the foreground. For instance, if a dog is detected in a large field, the resulting patch would represent the background of the field rather than the dog itself.
3. In the THEORETICAL ANALYSIS section, the derivation of formulas starting from the Generalization Bound needs further clarification, since the major concern of the proposed method is not the generalization problem. Additionally, the proposed method does not seem to be connected to the mathematical derivations presented.
4. The experiments do not compare the proposed method with latest state-of-the-art approaches in 2024.
5. The proposed method requires experimental validation of the effectiveness of the variance of radiation points.
6. The related work section lacks a review of recent studies from the past few years, with most of the cited works dating prior to 2021.
7. In Section 3.1 (Importance Prior Calculation Method), the meanings of the symbols used are not explicitly defined or explained.
8. In Section 3.2 (Optimization and Learning Strategy), there is a lack of detailed information about the "localization loss," which is crucial for understanding the proposed approach.
9. The methodology section lacks a clear overview of the entire process, including key details regarding training and inference stages of the model.
10. In the Implementation Details section, there is no information provided on the baselines used, the type of GPUs utilized, the learning rate applied during training and so on.
11. The experimental results do not include a comprehensive analysis of the impact of hyperparameters δ and k, which could be significant for the findings.
12. Figure 5 presents "Occlusion Ratios," but the term is not clearly defined or explained within the text, leaving the interpretation ambiguous.

**Questions:**

Please see the weaknesses.

---

### Official Review · Reviewer_QbYX · 2024-11-04

**Soundness:** 2
**Presentation:** 1
**Contribution:** 2
**Rating:** 3
**Confidence:** 3

**Summary:**

The paper proposes an adaptive masking strategy for low-shot training of visual grounding. The mask is generated by thresholding an importance prior computed from an attention module and Gaussian radiance fields. The paper provides both theoretical and empirical evidences that suggest the proposed masking strategy is effective in low-shot visual grounding.

**Strengths:**

The paper proposes a novel masking strategy by combining an attention-based prior and Gaussian radiance fields.
Both component have empirically shown to improve upon random masking, yielding significant improvement in closed-set, zero-shot, and low-shot visual grounding setting.

**Weaknesses:**

My major concerns are about the positioning of the paper and lacking baselines.

* While the paper focuses on low-shot visual grounding, I am curious about the broader impact and relevance of this direction.
Given the availability of numerous large-scale pre-trained vision-language models that have demonstrated strong generalization to unseen domains or categories (e.g., zero-shot and few-shot settings), I wonder if training a vision-language model from scratch with a limited dataset offers significant advantages in practice.
In particular, datasets with everyday objects (like COCO, used in the experiments) can be easily scaled up through methods such as web crawling.

* In the experiments, the primary comparison is made with a random masking baseline.
Considering there are several related works on adaptive masking (e.g., [1, 2, 3]), I believe it would be beneficial to compare the proposed approach with additional, more relevant baselines.
Such comparisons would help highlight the effectiveness of the specific adaptive masking strategy proposed, and discussing these works in the related work section would also provide stronger context for the contributions of this paper.

I also have some minor concerns:

* The loss terms ($L_{contrastive}, L_{localization}$) are not clearly defined.
Furthermore, there seems to be inconsistency in the naming of loss terms, such as $L_{mask}$ and $L_{IMAGE}$ mentioned in Section 3.3. Clarifying these terms would improve the clarity of the paper.

* The discussion in Section 3.3 feels somewhat underdeveloped. Does Theorem 1 suggest that the proposed adaptive masking strategy is more data-efficient than random masking? Additionally, how does this theoretical result align with the empirical findings?

* The use of the term "few-shot" throughout the paper may be somewhat misleading, as the portion of data used in the experiments (5-30%) appears larger than what is typically used in few-shot literature, which usually involves very small datasets (e.g., a few dozen examples).

References:

[1] Yang, Yifan, et al. "Attentive mask clip." Proceedings of the IEEE/CVF International Conference on Computer Vision. 2023.

[2] Wei, Zihao, Zixuan Pan, and Andrew Owens. "Efficient Vision-Language Pre-training by Cluster Masking." Proceedings of the IEEE/CVF Conference on Computer Vision and Pattern Recognition. 2024.

[3] Liang, Mingliang, and Martha Larson. "Centered Masking for Language-Image Pre-training." Joint European Conference on Machine Learning and Knowledge Discovery in Databases. Cham: Springer Nature Switzerland, 2024.

**Questions:**

Please refer to the weakness section.

---

### Note · Authors · 2024-12-08

I have read and agree with the venue's withdrawal policy on behalf of myself and my co-authors.